# The use of technical replication for detection of low-level somatic mutations in next-generation sequencing

Junho Kim[1], Dachan Kim[1], Jae Seok Lim[2], Ju Heon Maeng[1], Hyeonju Son[1], Hoon-Chul Kang[3], Hojung Nam[4], Jeong Ho Lee[2] & Sangwoo Kim [1]

Accurate genome-wide detection of somatic mutations with low variant allele frequency (VAF, <1%) has proven difficult, for which generalized, scalable methods are lacking. Herein, we describe a new computational method, called RePlow, that we developed to detect low-VAF somatic mutations based on simple, library-level replicates for next-generation sequencing on any platform. Through joint analysis of replicates, RePlow is able to remove prevailing background errors in next-generation sequencing analysis, facilitating remarkable improvement in the detection accuracy for low-VAF somatic mutations (up to ~99% reduction in false positives). The method is validated in independent cancer panel and brain tissue sequencing data. Our study suggests a new paradigm with which to exploit an overwhelming abundance of sequencing data for accurate variant detection.

[1] Department of Biomedical Systems Informatics and Brain Korea 21 PLUS Project for Medical Science, Yonsei University College of Medicine, Seoul 03722, South Korea. [2] Graduate School of Medical Science and Engineering, KAIST, Daejeon 34141, South Korea. [3] Department of Pediatrics, Division of Pediatric Neurology, Pediatric Epilepsy Clinics, Severance Children's Hospital, Epilepsy Research Institute, Yonsei University College of Medicine, Seoul 03722, South Korea. [4] School of Electrical Engineering and Computer Science, Gwangju Institute of Science and Technology, Gwangju 61005, South Korea. Correspondence and requests for materials should be addressed to J.H.L. (email: jhlee4246@kaist.ac.kr) or to S.K. (email: swkim@yuhs.ac)

Next-generation sequencing (NGS) has afforded research-ers the means with which to investigate somatic variants with tremendous accuracy. For many years, the usefulness of NGS was highlighted in cancer research, wherein mutations are clonally expanded and shared by the majority of cancer cells, thereby providing a sufficient variant allele frequency (VAF) that can be detected in a sample. However, recent applications of genome analysis, such as in liquid biopsy[1], noninvasive pre-natal testing[2], somatic mosaicism[3], tumor subclones[4], and cell lineage tracing[5], are fraught with somatic single-nucleotide variants (SNVs) that exist at low VAF. Increasing evidence sup-ports the contribution of low-level SNVs to various noncancerous diseases[6–9]. Accurate detection of these SNVs may prove to be the key to further expanding the use of NGS in biomedical research.

Detection of low-VAF somatic mutations is a challenge in conventional NGS. Even at a high-read depth, NGS shows a rapid drop in detection accuracy of low-VAF somatic mutations[10–12]. Attempts to address this issue have mainly focused on modifying sequencing protocols, such as tagging unique molecular identifiers[13,14], generation of tandem-copies[15], adding DNA-repair enzymes[16], and selection of mutation-harboring sub-samples (e.g., single-cell sequencing[17]). The common aim of these methods is to enhance signal-to-noise ratios by amplifying mutation-driven variant alleles while discriminating erroneous alterations in nonmutation sites: the majority of these errors are believed to originate from external DNA damage[18,19], which has been found to pervasively confound variant identification in genome resequencing projects[16]. While technical advances that seek to reduce these errors are important, a more general and sustainable approach is required to accelerate practical applica-tion of conventional NGS data.

In science, one of the key processes through which to yield accurate and reliable data is a measurement of replicates. Unlike other biological experiments, however, NGS for variant detection has been granted an exemption from experimental replication, mostly due to costs and a lack of analysis methods[20]. As NGS is rapidly diminishing in cost, we suspect that the use of replication could provide a general, efficient, and widely applicable means by which to detect rare but biologically important somatic variants.

Here, we develop a new probabilistic model (named RePlow) that jointly analyzes library-level replicates for accurate detection of low-VAF somatic mutations. Importantly, the method is platform independent. Given sequencing data, RePlow infers patterns of background errors intrinsic to a data set. According to these inferred error profiles, variants are called by identifying mismatched alleles for all replicates simultaneously. Compared to a single-sample-based variant calling, RePlow shows marked improvement in both sensitivity and specificity. Furthermore, we are able to confirm the accuracy of our model in independent cancer panels and to discover low-VAF variants (~0.5%) that were not detected with conventional variant calling settings. Our model demonstrates that exploiting replicates can be a cost-effective, scalable, and sustainable solution for detecting low-level somatic mutations, which has continued to remain elusive.

## Results

### The current state of detecting low-VAF somatic mutations.
First, we sought to examine the bona fide accuracy of current conventional NGS techniques and algorithms in calling low-VAF somatic mutations. We prepared a test-base data set for the measurement (Fig. 1a). Unlike in silico simulations, directly pooled genomic materials reflect the variety of errors across the entire sequencing step. Thus, genomic DNA from two indepen-dent blood samples was mixed to mimic somatic mutations at

four different VAFs: 0.5, 1, 5, and 10% (designated as samples A–D, respectively). Sequencing of the material provided a set of control positives (645 true variants) and negatives (66,485 non-variant sites) for determining detection accuracy, including sen-sitivity and false-positive rate (FPR). The test-base data set consisted of library- and sequencing-level replicates for three distinct platforms: hybridization-capture-based Illumina sequen-cing (ILH, up to 1000×) and amplicon-based Illumina and Ion Torrent sequencing (ILA and ITA, respectively, up to 10,000×) (see Methods). The sequencing data sets were further downsampled by an interval of 100× (ILH) or 1000× (ILA and ITA) to investigate the effect of read depths (Supplementary Table 1). The feasibility of detecting low-VAF variants with eight somatic variant callers was evaluated on the test-base data[21–27]. In their near-default settings (disabled coverage limit), most callers lost most of the variants in samples A and B since the majority of them were not originally designed to detect such low levels (Supplementary Fig. 1). Parameter optimizations including tumor-cellularity (see Methods) enabled detection of the lost variants, however, it was also accompanied by a tremendous increase in false-positive calls (7–400 k per Mb, Fig. 1b and Supplementary Fig. 2). While increasing the read depth generally improved the sensitivity, it did not enhance the overall perfor-mance of the callers due to large increases in FPRs. We noted a remarkably higher FPR for ILA, in which targeted genomic regions are covered by a smaller number of amplicons, compared to ITA, thus requiring more polymerase chain reaction (PCR) cycles for library preparation to achieve the same depth: PCR generates DNA damage, leading to errors in sequencing. In such an environment, high-depth sequencing can even lower sensi-tivity (Fig. 1b left). We found that most of these false-negative calls in high-depth ILA were triallelic, caused by the accumulation of errors at true variant sites (Supplementary Fig. 3).

The allele frequency distributions of the true and false calls in the test-base data set confirmed the intractability of current forms of sequencing data analysis. We found a consistent level of background errors (1–3%) in all three platforms (higher in amplicon sequencing) that dominated true signals at a VAF of ≤1% (Fig. 1c). Accordingly, additional filtering with a hard VAF cut-off value was unable to separate erroneous variants. Likewise, the distributions of probabilistic odd-ratio scores ($LOD_T$ score[23]) severely overlapped between mutations and errors (Fig. 1c right). Moreover, except read-pair orientation bias in ILH, none of the commonly used features for variant filtration, such as base call quality, mapping quality, number of per-read mismatches, and indel proximity, or common postfiltering steps were able to mitigate the problem (Supplementary Figs. 4 and 5). These results refute previous perspectives that suggest errors can be distinguished from true low-level mutations through stringent filters[16,20].

### Using replicates: primitive models.
The NGS sequencing process can be divided into three steps: (1) sample preparation, (2) library preparation, and (3) sequencing, each of which can generate genuine errors (Fig. 2a). For example, sample contamination, PCR amplification error, and overlapping fluorescence signals are frequently observed errors in each respective step[20]. Technical replication aims to measure the variance of these errors between data sets. However, said measurement is limited by the time of the replication, because errors generated in preceding steps will be shared in all following replicates. Thus, repeated sequencing of the same libraries or a collection of sequencing reads does not provide any information concerning PCR errors or DNA damage. Accordingly, we referred to library-level replicates as proper technical replicates in this study.

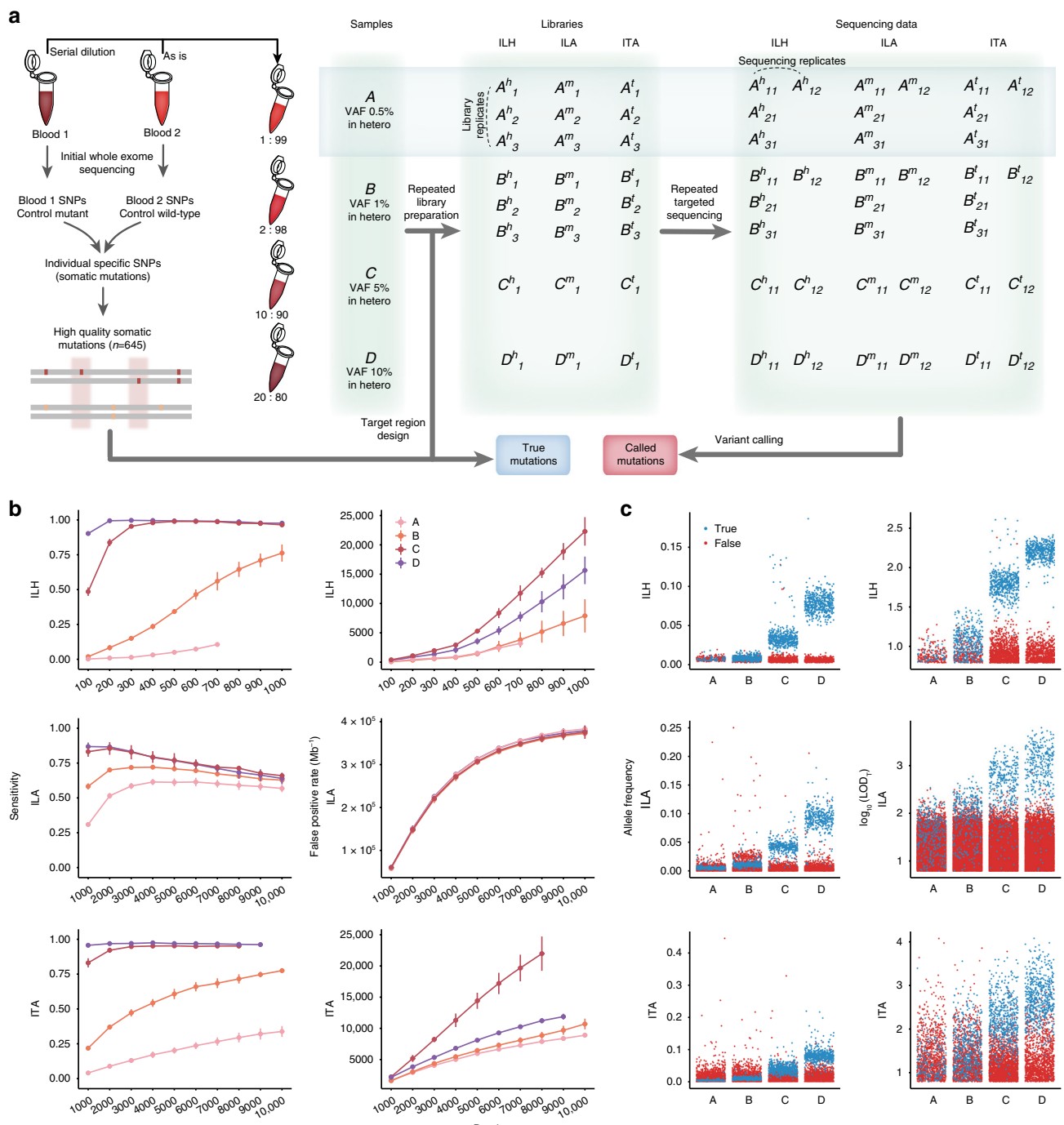

**Fig. 1** Assessment of conventional algorithms for detecting mutations with low-allele frequency. **a** Schematic of experimental design for test-base sequencing data. Four distinct sample mixtures (A, B, C, and D) were prepared and sequenced with three different sequencing platforms (ILH, ILA, and ITA). Constructed libraries from each platform were sequenced twice to produce sequencing replicates ($X_{11}$ and $X_{12}$). For samples A and B, two independent sets of sequencing library were additionally prepared to sequence data from library replicates ($X_{21}$ and $X_{31}$). Each set of sequencing data was sequentially downsampled ten times to evaluate the effects of read depth. All generated datasets were analyzed, and average performances were reported for each depth and platform. **b** Sensitivity and FPR of conventional methods (MuTect with adjusted parameters, others in Supplementary Figs. 1 and 2) by sequencing depth and VAF for each sequencing platform. Points are depicted within the maximum depth of the sequencing data (Supplementary Table 1). Error bars, 95% confidence intervals. Source data are provided as a Source Data file. **c** Distribution of allele frequencies and probabilistic odd-ratio scores ($LOD_T$) for true-positive and false-positive calls for each sample mixture (colored by blue and red, respectively). ILH hybrid-capture-based Illumina sequencing, ILA amplicon-based Illumina sequencing, ITA amplicon-based Ion Torrent sequencing, VAF variant allele frequency, FPR false-positive rate

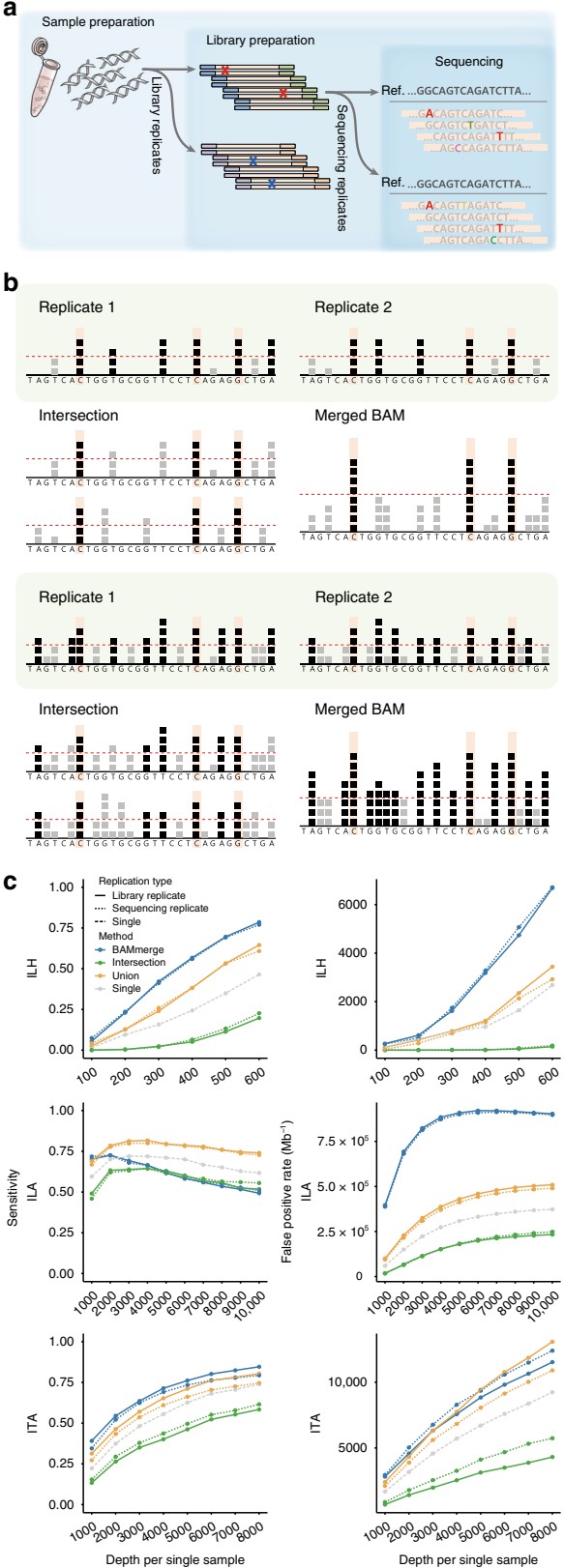

**Fig. 2** Use of replicates with primitive models. **a** Experimental steps in the typical NGS process. Errors can be generated at each step. Note that background errors in the library preparation step (red marks and bases) cannot be discriminated with the sequencing replicates (pseudo-replicates). **b** Description of primitive approaches (intersection and BAM-merge) with their expected (upper) and real (lower) effects. Each square represents an observed B allele for a given position. Positions with a number of B alleles beyond the detection threshold (red dashed line) are called as mutation candidates (positions with black squares). Both approaches are expected to discriminate true variants (orange-shaded positions) from false calls based on the randomness of error (upper). However, in real high-depth data, both approaches are ineffective due to excessive background errors (lower). **c** Sensitivity and FPR of the primitive approaches with sample B (1% VAF) for each platform. Primitive approaches were applied for both library (solid lines) and sequencing (dotted lines) duplicates. Calls from the single sample (dashed lines) are also depicted to evaluate the improvement with replicates. All mutation calls were made by MuTect. Source data are provided as a Source Data file

seen as the conventional "call-and-validate" strategy, where initial candidate variants undergo independent validation in subsequent data sets. In the BAM-merge model, alignment files (BAM files) from all replicates are merged to a single file and fed into a caller. In both models, the expected benefits rely on an assumption of error randomness: ideally, true mutation signals will accumulate, and errors will be dispersed (Fig. 2b upper).

In the present study, we found that both models only provided mediocre improvement above conventional single-sample-based variant calling for the low-level mutation detection (Fig. 2c). The intersection model substantially lowered sensitivity, as was expected. Moreover, it failed to effectively reduce FPRs in two amplicon-based platforms (ILA and ITA in Fig. 2c green lines, and Supplementary Fig. 6). While the BAM-merge model substantially increased sensitivity in two platforms (ILH and ITA), it generated a troubling number of false positives in all platforms (blue lines, Fig. 2c and Supplementary Fig. 6). Unlike the ideal condition, we noted two critical factors as sources of these inefficiencies (Fig. 2b lower). First, the overall amount of background errors was higher than expected, affecting a wide range of genomic positions. This resulted that replicates of nonvariant sites were being called as true variants (Supplementary Fig. 7). Second, merging BAM files increased the read depth and lowered the VAF threshold with which to achieve caller's probabilistic significance, promoting false positives (Fig. 2b, lower right). The combined effects of background errors and a higher read depth elicited extremely complex model behavior: for example, the FPR in the BAM-merge model for ILA starts to decrease from 6000× coverage (Fig. 2c middle right panel), as more than two different erroneous alleles begin to accumulate to form a triallelic site in the false positive sites (Supplementary Fig. 8).

We confirmed that background errors cannot be separated with mere use of replication. Additionally, the results showed that the primitive models do not differentiate library-level replicates from those at the sequencing level (Fig. 2c, dotted lines), which implies the improper use of technical replicates. Therefore, more sophisticated approaches are needed to overcome these challenges.

**Using replicates: the RePlow model**. To make better use of replication in NGS data analysis, we developed RePlow, a new model that jointly analyzes library-level replicates to call low-VAF somatic mutations in a data set. In an attempt to address the

In the absence of systematic methods, two primitive approaches can be implemented to test the effect of replication on detecting low-VAF variants: intersection and BAM-merge (Fig. 2b and Methods). The intersection model is based on the reproducibility of variants. Thus, only variants that are called in every replicate are finally reported. The intersection model can be

challenges stated above, our primary goal for the method was to achieve robust discrimination of background errors based on repeated observations. To achieve the goal, we designed the model to infer a probability distribution of background errors in each replicate, relying on the given raw data ("on-the-fly manner"), so as not to lose generality. Final variant calling is conducted according to merged probabilities among replicates, reflecting the concordance of mismatched allele compositions.

For the on-the-fly error profiling, we devised a strategy that measures the amount of mismatched alleles in alignment caused by background errors, the mismatch overrepresentation score (MOS, see Methods). In conventional models, the cause of mismatches is regarded as either (1) the presence of variants or (2) sequencing errors in uniquely mapped regions. The expected amount of sequencing errors can be calculated from the collection of base-call quality scores in mapped reads. Thereby, an unexpectedly large number of mismatches (e.g., high VAF in high-quality reads) is directly interpreted as the presence of true variants. In designing RePlow, we considered background errors as additional causes for mismatches that distort the distribution of mismatches but are not recognized by the base-call quality. Briefly, MOS scores are calculated by the discrepancy between the expected and the observed amount of mismatches to construct the probability distribution functions (PDFs) of a background error-induced VAF. We presumed that sampling MOS scores in a large number of nonvariant sites (e.g., >10,000 base pairs) with high-quality alignment scores could profile sample-specific background errors (see Methods).

Calculation of MOS scores on a matched control sample (variant-free) in the test-base data set identified patterns and the levels of background errors generated for the three platforms, each of which showed a genuine signature (Fig. 3a). The overall error levels were higher for the amplicon-based platforms (ILA and ITA) than the hybrid-capture platform (ILH). Additionally, the error levels were specific to the sequence context. We noted excessive background errors in $A > G$ ($T > C$) and $C > T$ ($G > A$) transitions for ILA, which is considered a signature of PCR error during library amplification[28]. Meanwhile, ILH data contained a higher level of $C > A$ ($G > T$) substitutions, a well-known artifact caused by DNA oxidation during the hybrid-capture specific sonication process[19]. The inferred context-specific background error profiles were also consistent with the biased patterns of observed VAFs at nonmutant sites in the test-base data set (Fig. 3b, red dots), which supports that the major source of erroneous variant calling comes from intractable background errors, not controllable base-call errors (e.g., by removing low-quality reads). Using the distribution of MOS scores from every sampled position, we attempted to construct PDFs for two random variables: (i) VAFs acquired as background errors and (ii) the sequence context (Fig. 3c). Since no canonical probability distribution is known for NGS background errors, we drew the empirical cumulative distributions for each substitution type and fit them to an exponential distribution (see Methods). In doing so, we confirmed that the inferred distributions (red lines) closely approximated their true distributions (black lines).

The VAF distribution of the test-base samples (Fig. 3b) provided two important justifications for using replicates: (1) true and false mutations become more separable in a higher dimension (in contrast to a single data set, Fig. 1c), and (2) VAFs of true variants are more concordant in replicates (Kolmogorov–Smirnov (KS) test, $P < 2.2 \times 10^{-16}$, $P = 9.2 \times 10^{-13}$, $P = 6.7 \times 10^{-9}$ for ILH, ILA, and ITA, respectively, Supplementary Fig. 9). RePlow implements a probabilistic model with which to quantify these two features, based on a general

number of replicates (Fig. 3d). Briefly, RePlow calculates the probabilities of being a true variant and an error for a given position in every replicate. The probability of error is estimated by the inferred sample-specific PDF, while the probability of variant is estimated by binomial approximation with the averaged VAFs, that evaluates the concordance between replicates, to give more advantages to the candidates with concordant VAFs (see Methods). Both probabilities are jointly analyzed to estimate the likelihood of a true variant in a sequence context (Fig. 3d). Sites that have a higher probability being a variant than an error are then treated with postfilters to eliminate systematic errors that are not captured by the error model (Supplementary Methods). Passing sites are finally considered as variant candidates.

**Variant detection with RePlow.** We tested RePlow on the test-base data to compare its performance with single and the primitive replication models (Figs. 4a, 1% VAF is shown). Note that a common RePlow model was applied to the three different platforms (ILH, ILA, and ITA) without any platform-specific adjustments. The most prominent improvement achieved with RePlow was a remarkable drop in FPRs (298.1, 6069.6, and 329.8 Mb$^{-1}$ for ILH, ILA, and ITA, respectively, red lines in Fig. 4a), reductions of 70.2, 97.5, and 94.4% compared to the most precise primitive model (intersection, green lines) and reductions of 96.2, 98.4, and 96.9% compared to the single-sample calling. Moreover, the reductions in FPR were achieved without a loss of sensitivity, comparable to that with union or BAM-merge modeling (orange and blue lines). These overall improvements led to outstanding performance for RePlow in a balanced measure (F-score). Similarly, RePlow achieved the highest performance at a lower VAF (sample A, 0.5%) and with the same total sequencing throughput (half-coverage for replicates compared to the single library) for a minimum read depth of >400× per replicate (Supplementary Figs. 10 and 11). Application to triplicates increased the model accuracy even more, especially in terms of sensitivity, although there was a decrease in precision for ILA (Supplementary Figs. 12 and 13).

Cross-platform replication is a widely used validation method (e.g., initial calling in ILH and validation in ILA). Being platform independent, RePlow can be applied to any combination of replicates generated by multiple platforms. Accordingly, we sought to test the bona fide effects of such validation scenarios of the three platforms in pairs and in comparison to RePlow (Fig. 4b). Although high-precision supports the reliability of cross-platform validated variants, a significant loss of true low-VAF mutations was observed (sensitivities of 47.1, 57.8, and 49.3% for ILH × ILA, ILH × ITA, and ILA × ITA pairs, respectively). Meanwhile, we found that joint analysis of replicates using RePlow was superior to the cross-platform validation approach, increasing sensitivities by more than 20% (75.8, 78.5, and 68.0% in the same order of pairs) while maintaining high precision (Fig. 4b, red vs. green bars). These results indicated that many low-level mutations are falsely rejected by the current validation method, a substantial portion of which can be rescued by RePlow.

Next, we applied RePlow to an independent dataset. A commercial reference standard with 35 cancer hotspot SNVs with VAFs of 1.0–1.3% was prepared and sequenced using two widely used cancer panels (Illumina SureSelect and Ion AmpliSeq-based, see Methods for details) in up to triplicates (Fig. 4c and Supplementary Table 2). We found that both the single and the primitive replication models failed to detect ~10 true mutations, especially those at triallelic sites, most of which were successfully called by RePlow. The FPRs of the conventional models varied across platforms, from 600 to 60,000 Mb$^{-1}$ (higher in Ion AmpliSeq). However, RePlow showed reduced FPRs of

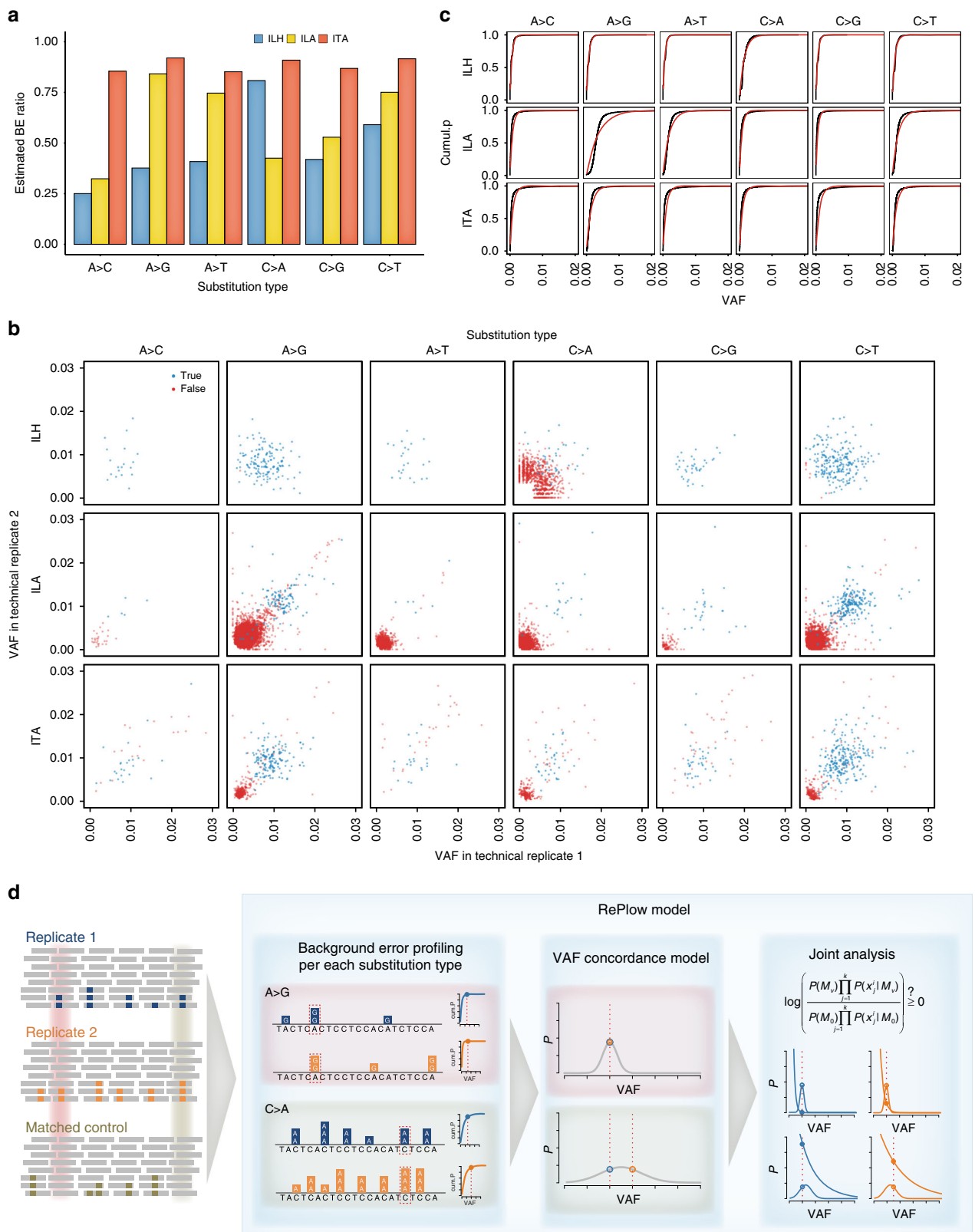

180–250 Mb$^{-1}$, including a 99.24% reduction in FPR for Ion AmpliSeq. These results confirmed the general applicability of RePlow. As false positivity in clinical multigene tests is as devastating as false negativity, library-level replication can be considered as an efficient approach, providing a drastic gain in specificity with only a relatively small increase in cost.

Finally, we applied RePlow to real disease data, attempting to identify disease-associated or -causing somatic mutations with low-allelic frequency that might have been missed in a singleton of deep sequencing. Recently, childhood intractable epilepsy with focal cortical dysplasia or low-grade tumor (e.g., ganglioglioma) has been reported as being caused by low-level somatic mutations

**Fig. 3** Development of the RePlow model. **a** The estimated proportion of background errors (BEs) from total mismatches by substitution type. MOS values were measured for each substitution type from total mismatches of matched control samples. Positions with germline variants were excluded to assume that all mismatches originated from either sequencing or background errors. The ratio of the sum of MOS scores to the total mismatch count is regarded as an estimate of the BE proportion. **b** VAF distribution of called mutation candidates from library replicates of sample B (1% VAF) for each platform. All candidates were called by MuTect in at least one replicate. True positive and false-positive calls are colored in blue and red, respectively. **c** Empirical and fitted cumulative distribution for the VAFs of background errors. To estimate the PDF of background errors, VAF profiles based on the MOS value of each position (empirical cumulative distribution, black lines) were constructed and fitted by cumulative exponential distribution (red lines) (see Methods). PDFs were then constructed for each substitution type with the estimated parameter of the cumulative exponential model. **d** Overview and examples of mutation detection by RePlow. Mapped sequencing data of replicates and matched control are taken as input. For each data set, VAF profiles of background errors per substitution type are constructed first to estimate the PDF. Then, each genomic position is analyzed to calculate probabilities of being a variant or an error using estimated concordance models with the average VAF (normal distribution) and background error profiles (exponential distribution), respectively (see Methods). Both probabilities are jointly analyzed to estimate the likelihood thereof in a sequence context. Sites with a C > A mutation (green-shaded area) show a higher VAF than A > G mutation sites (red-shaded area). However, due to the excessive occurrence of context-specific error (C > A) and VAF discordance between replicates, RePlow selects only the A > G mutation site as a final candidate based on the joint analysis result. MOS mismatch over-representation score, PDF probability density function

in *MTOR* or *BRAF*[3,29,30]. Importantly, a somatic mutational burden of even ~1% in the focal brain has been deemed sufficient to cause intractable epilepsy[3,31]. We obtained specimens from two intractable epilepsy patients with matched brain-peripheral (e.g., blood or saliva) tissue found to be negative for any pathogenic mutations in a singleton of deep targeted sequencing of *MTOR*, *BRAF*, and other related genes. We performed two additional replications in brain tissues from these mutation-negative patients. In result, a total of seven mutation candidates were called by RePlow, compared to only one candidate called by the intersection method that did not overlap at all. Among the seven candidates called by RePlow, novel missense mutations in *MTOR* (p.S2215F) and *BRAF* (p.V600E) from each sample were previously reported as disease-associated or -causing mutations[3,29,32–35]. All seven mutations were tested with droplet digital PCR; five candidates including both missense mutations were successfully validated (Fig. 4d and Supplementary Fig. 14) and designing primers for the remaining two candidates failed. The two missense mutations showed extremely low VAFs in sequencing data (0.77 and 0.45% on average), which were not able to achieve high enough significance levels to be detected by previous methods. Taken together, these results support the use of our RePlow model allow for more accurate detection of low-level somatic mutations with far less number of false positives via replication of conventional NGS.

## Discussion

Rapid advances in DNA sequencing technology have helped reduce the costs of sequencing at a rate that outpaces Moore's law. For the last 10 years, sequencing costs have declined by a factor of ~10,000, a trend that is expected to continue. Every 1 year, researchers can generate sequencing data at a 1.5–4 times higher throughput at the same cost. Overall, the general consensus is that the cost of sequencing itself will no longer act as a bottleneck to genome research in the near future[36]. Nevertheless, merely increasing the numbers of a sample is not an ultimate solution to attaining research objectives, such as discovery of cancer driver mutations, which is already approaching a plateau[37]. At this point, we need to scrutinize the directions to which lower costs for DNA sequencing provide actual research benefits beyond sample size or read depth. We suggest that replication will prove useful to traversing the current limits of variant detection, facilitating robust identification of low-level somatic mutations.

We would like to note that the use of replication is not substitutive for or mutually exclusive with other technologies for detecting low-level somatic mutations. What we have described in this study is not merely a specific method, moreso a paradigm for adding a new dimension to the current methods of mutation

calling by which previously unquantifiable background errors in presequencing steps can be profiled and undergo technical replication. As we have shown, RePlow can be applied to multiple platforms of completely different data characteristics without the need for a priori background error profiles. By virtue of its platform independence, RePlow should maintain the ability to improve the performance of mutation calling for upcoming sequencing technologies through replication. Moreover, the possibility remains for wider application to nonconventional sequencing (e.g., single-cell sequencing and barcoded sequencing) or to biological replicates (multiple samples from the same individual, e.g., different tumor biopsy portions) for detecting shared mutations with very low-allelic fractions, which would be helpful in detecting a second-hit mutations in tumors or to analyze the lineage and clonality. Such advances will provide us more rigorous and extensive testing of the influence of low-level mutations.

## Methods

**Construction of a spike-in, test-base genome**. Spike-in, test-base data were prepared by mixing genomic DNA from two independent blood samples. Subsets of unique germline single-nucleotide polymorphisms (SNPs) from one sample (reference alleles in another) served as an answer set of somatic mutations that have been found by variant callers. We first generated whole exome sequencing data (WES, ~400× coverage) of each sample to verify their genotypes and to select unique variants. Since we focused on measuring the accuracy of mutation callers according to changes in VAF, we only considered genomic regions with high-mapping quality (average mapping quality ≥ 58 assessed by WES data of 1000 Genomes Project) to minimize the influence of mapping ambiguity for variant calling. We also restricted target regions to satisfy all following conditions for both samples to avoid the influence of other biases: (i) exons containing unique variants with a read depth ≥ 20, (ii) indel-free exons, and (iii) exons containing clipped reads < 5. From those regions, only exons with unique heterozygous SNPs were selected to control the overall consistency of VAFs. A total of 645 unique variants from 564 exons were selected as an answer set for the spike-in data. To mimic mutations with different allele frequencies, four distinct mixtures were made by diluting samples at ratios of 0.01, 0.02, 0.1, and 0.2 to represent mutations with VAFs of 0.5, 1, 5, and 10%. The samples were named A–D, respectively (Fig. 1a). Intact blood gDNA from wild type sample served as a matched control. All human tissues including blood samples for test-base data and brain tissues of intractable epilepsy patients were obtained with informed consent in accordance with protocols approved by the Severance Hospital and KAIST Institutional Review Board and Committee on Human Research.

**Sequencing of test-base genome**. Two different sequencing platforms, Illumina and Ion Torrent, were used to generate sequencing data for all mixture samples. For the Illumina platform, both hybridization-capture and amplicon-based sequencing were performed to compare the results of each library preparation method. As a result, all mixture data were sequenced through three different platforms: ILH, ILA, and ITA. Agilent Sure Design online tools (https://earray. chem.agilent.com/suredesign/), Illumina design studio (https://designstudio. illumina.com), and Ion AmpliSeq Designer (https://ampliseq.com/) were used to design custom probes that cover selected mutations for each sequencing platform. Since the target coverage of the designed probes differed for each platform,

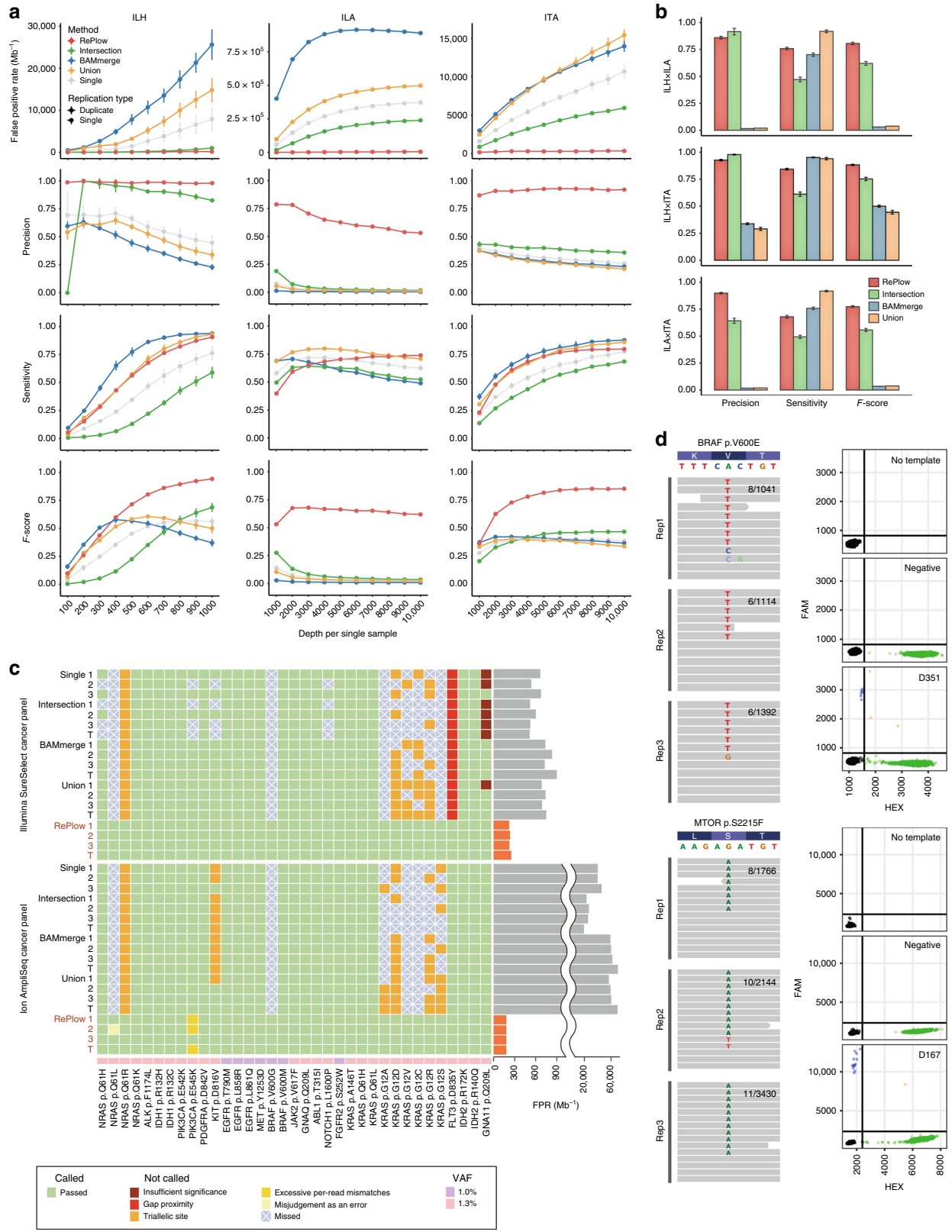

platform-specific answer sets were made that contained 513, 540, and 591 variants for ILH, ILA, and ITA, respectively. For ILH, samples were processed with target capture and library preparation according to Agilent's protocol and sequenced on an Illumina HiSeq 2000 sequencer (101 × 2 bp read length, ~1000× coverage). Samples with ILA followed the TruSeq preparation protocol and were sequenced on an Illumina HiSeq 2500 sequencer (151 × 2 bp read length, ~10,000× coverage). Ion Ampliseq protocol was carried out for ITA with an Ion Proton sequencer (125–275 bp amplicon range, ~10,000× coverage). All samples were sequenced

twice from each constructed library; we called these data pairs as sequencing replicates ($X_{11}$ and $X_{12}$). For samples A and B (0.5 and 1% VAF), two library replications (independent preparation of sequencing library from the same DNA sample, $X_{21}$ and $X_{31}$) were additionally performed for all sequencing platforms to compare differences in sequencing and library replication. All generated data were downsampled 10 times (100× to 1000× for ILH and 1000× to 10,000× for ILA and ITA) to track the detecting accuracy by sequencing depth. Details on the sequencing data and preprocessing procedures are described in Supplementary

**Fig. 4** Comparative performances of RePlow and the primitive approaches. **a** Performance assessment with the library replicates of test-base data. FPR, precision, sensitivity, and *F*-score were measured for sample B (1% VAF). All three combinations of duplicates were tested, and their average performances were reported with 95% confidence intervals (typically smaller than marks). **b** Performance assessment with the combination of replicates in multiplatforms. All pairs of library replicates between different platforms were tested with test-base data of sample B. Only the data sets with the highest depth of each platform were used for the combination (1000× for ILH and 10,000× for ILA and ITA). Error bars, 95% confidence intervals. **c** Independent assessment with a reference material sequenced by two widely-used cancer panels. Detection of 35 true cancer hotspot SNVs (1–1.3% VAF) were tested for all combinations of library triplicates ($X_1X_2$, $X_2X_3$, $X_1X_3$, and $X_1X_2X_3$ are denoted as 1, 2, 3, and T, respectively). Green shading means a correct detection, and other colors represent the reason for the rejection or no detection (with X marks). FPRs of RePlow are highlighted in orange to emphasize their reductions therein, compared to other primitive approaches. **d** Experimental validation of rescued low-level mutations from the samples negative for pathogenic mutations in previous analysis. Observed allele counts are described in each replicate (left). Droplet digital PCR results for no DNA template (No template), DNA from healthy controls (negative), and disease samples are shown together for each site (right). Green and blue dots represent wild type- and mutant-specific signals, respectively. Source data for **a**, **b** are provided as a Source Data file

Table 1 and Supplementary Methods. Information for designed target regions per sequencing platform and platform-specific answer sets are listed in Supplementary Datas 1 and 2.

**Performance assessment of conventional algorithms**. We attempted to measure the conventional performance of variant calling, especially for variants with low frequency (≤1%). All generated data were analyzed by eight variant callers: MuTect[23], VarScan2[21], Strelka[22], VarDict[24], FreeBayes, Virmid[25], LoFreq[26], and deepSNV[27]. Default parameter settings were tested first for each method with minimal adjustments that disable the coverage limit. Since most algorithms were not carefully designed for high-depth and amplicon-based sequencing data, calls with default parameters lost most true mutations with low frequency for all platforms (Supplementary Fig. 1). Thus, we adjusted parameters to recover those variant calls. Besides each caller-specific options, three criteria were commonly applied: (1) the exact value of expected VAF (0.5, 1, 5, and 10% for sample A–D, respectively) was provided if a caller accepts this information. (2) If it is applicable, minimum VAF to be called was set to 0.25% (a half of true VAF from sample A) to remove low-level background errors without affecting true mutation detection from sample A. (3) For amplicon datasets, filters for strand bias and clustered positions were disabled due to the nature of their generation. Through these adjustments, mutations at all mixture levels were successfully discovered by most callers but could not avoid the dramatic increase of false positive rate (Supplementary Fig. 2). Detailed information for parameter adjustment is described in Supplementary Methods.

**RePlow model for variant detection**. Like other conventional callers, RePlow basically detects somatic variants by comparing the probability of two alternative models: a variant model ($M_v$) and a reference model ($M_0$) that treats all mismatches as error calls. The major difference in RePlow is that replicated data are considered simultaneously for the calculation of probabilities. For a genomic position $i$ with $k$ replicates, we denote the total number of reads (sequencing depth) of replicate $j$ as $n^i_j$ and the number of reads with variant alleles as $b^i_j$. VAFs of the $i$th position from replicate $j$, $x^i_j$, can be calculated as $b^i_j/n^i_j$. Given observations from position $i$, the log ratio ($S^i$) of probability for both models is defined as:

$$S^i = \log\left(\frac{\prod_{j=1}^{k} P\left(M_v \middle| x^i_j\right)}{\prod_{j=1}^{k} P\left(M_0 \middle| x^i_j\right)}\right) = \log\left(\frac{\prod_{j=1}^{k} P(M_v)P\left(x^i_j \middle| M_v\right)}{\prod_{j=1}^{k} P(M_0)P\left(x^i_j \middle| M_0\right)}\right). \quad (1)$$

We assume that all replicates share the identical set of true mutations. Based on this assumption, if the $i$th position of one replicate is assumed to be mutated, the rest should also be mutated thus their prior probabilities will be 1. Therefore, prior probability will be applied once, regardless of the number of replicates, and Eq. (1) can be rewritten as follows:

$$S^i = \log\left(\frac{P(M_v)\prod_{j=1}^{k} P\left(x^i_j \middle| M_v\right)}{P(M_0)\prod_{j=1}^{k} P\left(x^i_j \middle| M_0\right)}\right). \quad (2)$$

Since $P(M_v)$ and $P(M_0)$ are prior probabilities of being mutated or not for a given position, summation of those values should be 1. If we can estimate each value of $S_i - P(M_v)$, $P(x^i_j|M_v)$, $P(M_0)$, and $P(x^i_j|M_0)$—for all genomic sites, variant candidates can be determined by selecting genomic positions with $S^i > 0$. In other words, observed positions that are more likely to be explained by somatic variants than error calls will be selected as variant candidates. To achieve this, we carefully

designed both probability models $M_v$ and $M_0$ with unique features that have not been considered by conventional callers.

**Error model ($M_0$) estimation**. The reference model $M_0$ supposes that all mismatches at a given position are generated by errors. Previous methods have generally used the base quality score of an observed mismatch to estimate the probability that the mismatch represents an error. However, we argue that such an approach is insufficient to accurately determine this probability, depending on the type of error involved. We found that in addition to sequencing errors, a second type of error exists in those mismatches, which we refer to as background errors, the two generally occurring at different experimental steps: the former at the sequencing step and the latter at the library preparation step (Fig. 2a). Since mismatches caused by background error do not affect their base call quality, most previous callers continue to generate false-positive calls at positions with background errors, all of which have high base quality scores. Thus, we designed a new model to enable estimation of the probability that mismatches remaining after accounting for sequencing errors are background errors. We first constructed the adjusted VAF profiles that represent background errors only; we did this by subtracting the portion that can be explained by sequencing errors from all mismatches, which portion had previously been determined using base call quality scores. We then fit a parametric distribution to utilize corresponding PDFs to calculate the probability of being background errors.

To construct adjusted VAF profiles of background errors, we first collected all genomic positions that possess at least one nonreference alleles and estimated the expected count of background errors for each site. Positions that were called by GATK or commonly called by MuTect for all replicates were excluded, since there is a high probability of them being actual germline or somatic variants. By definition, the expected count of a sequencing error can be inferred from the base quality scores of variant alleles. Denoting the base quality score (Phred-scale) of read $l$ at the genomic position $i$ by $q^i_l$, the expected count of sequencing errors for replicate $j$, $b_{SE}{}^i_j$, can be calculated as $\sum_{l=1}^{b^i_j} 10^{-\frac{q^i_l}{10}}$. Then, we estimated the expected count of background error $b_{BE}{}^i_j$ by subtracting $b_{SE}{}^i_j$ from the number of reads with the variant allele $b^i_j$ based on our assumption, which is that a given mismatch from a nonvariant site should either be from a sequencing or background error. We defined this discrepancy as the MOS, which represents an unexplained number of mismatches by base-call quality (sequencing error).

$$\text{MOS}^i_j = b_{BE}{}^i_j = b^i_j - b_{SE}{}^i_j = b^i_j - \sum_{l=1}^{b^i_j} 10^{-\frac{q^i_l}{10}}. \quad (3)$$

We utilized the number of positions with $b_{BE}{}^i_j > 0$ as an empirical estimate of the number of positions that possesses mismatches caused by background errors. The ratio of positions with $b_{BE}{}^i_j > 0$ over the whole target region, $f_{BE}$, is thus considered as the estimated probability that a given position has a background error. The value of $f_{BE}$ is calculated from each replicate first and then averaged over replicates. The only parameter that has to be supplied by the user for the RePlow model is the probability that a given position has a somatic variant, $f_v$. The default value is provided as $3 \times 10^{-6}$, which is a typical mutation frequency commonly used in previous methods[23]. Since $P(M_v)$ and $P(M_0)$ in the variant detection model represent prior probabilities that a given mismatch-containing site is a mutation or an error, the relative ratio of $f_v$ and $f_{BE}$ are used as $P(M_v)$ and $P(M_0)$ for $S^i$ calculation.

$$P(M_v) = \phi_v = \frac{f_v}{f_v + f_{BE}}, \quad P(M_0) = \phi_{BE} = \frac{f_{BE}}{f_v + f_{BE}}, \quad \phi_v + \phi_{BE} = 1. \quad (4)$$

For every site with $b_{BE}{}^i_j > 0$, the adjusted VAF $x_{BE}{}^i_j$ is then calculated as $b_{BE}{}^i_j/n^i_j$ and is used to fit the distribution function. Exponential distribution is chosen to be fit, based on the observed shape of the empirical cumulative distribution function. Maximum-likelihood estimation for the parameter of exponential distribution $\lambda_{BE}$

is computed by the fitdistr package in R. Based on the estimated parameter, the likelihood of a given observation for the $M_0$ model can be calculated as:

$$P\left(x_{\mathrm{BE}\,j}^{\;i}|M_0\right) = \mathrm{Exp}\left(x_{\mathrm{BE}\,j}^{\;i}; \lambda_{\mathrm{BE}}\right). \quad (5)$$

Note that estimated parameter $\lambda_{\mathrm{BE}}$ has different values for each substitution type in a single-data set: depending on the sequencing platform, distributions of background errors differ greatly between substitution types (Fig. 3b). Therefore, RePlow separately performs error model estimation for each of the six substitution types (A > C, A > G, A > T, C > A, C > G, C > T) and uses the corresponding value of $\lambda_{\mathrm{BE}}$ for a given observation.

**Variant model ($M_v$) estimation.** The likelihood of $M_v$ is estimated using a binomial distribution. Since we assume that all replicates hold the same somatic variants, concordant VAFs are expected to be observed at the mutated site. On the other hand, errors would hardly show identical VAFs between replicates. We devised a model to reflect this VAF concordance to discriminate true variants from error calls. This model enables us to detect true mutations with the VAF less than those of background errors (Fig. 3d). The mean value of $x_{\mathrm{BE}\,j}^{\;i}$ for all replicates is used for the success probability of a binomial trial, giving a high probability only if concordant VAFs are observed between replicates.

$$\widehat{\mu}_{\mathrm{BE}}^{\,i} = \sum_{j=1}^{k} \frac{x_{\mathrm{BE}\,j}^{\;i}}{k}. \quad (6)$$

For each observation, $n_j^i$ and $b_{\mathrm{BE}\,j}^{\;i}$ are considered as the number of trials and the number of successes, respectively. Therefore, the probability of each observation can be calculated by a binomial distribution with the estimated success probability.

$$P\left(b_{\mathrm{BE}\,j}^{\;i}|M_v\right) = B\left(b_{\mathrm{BE}\,j}^{\;i}; n_j^i, \widehat{\mu}_{\mathrm{BE}}^{\,i}\right). \quad (7)$$

Since $b_{\mathrm{BE}\,j}^{\;i}$ can be a noninteger value, the likelihood is estimated through normal approximation of a binomial distribution $(B(n_j^i, \widehat{\mu}_{\mathrm{BE}}^{\,i}) \sim N(n_j^i \widehat{\mu}_{\mathrm{BE}}^{\,i}, n_j^i \widehat{\mu}_{\mathrm{BE}}^{\,i}(1 - \widehat{\mu}_{\mathrm{BE}}^{\,i})))$. Then, the observed VAF $x_{\mathrm{BE}\,j}^{\;i}$ will also follow a normal distribution, which is weighted by $1/n_j^i$ from the approximated distribution for $b_{\mathrm{BE}\,j}^{\;i}$:

$$x_{\mathrm{BE}\,j}^{\;i} \sim N\left(\widehat{\mu}_{\mathrm{BE}}^{\,i}, \frac{\widehat{\mu}_{\mathrm{BE}}^{\,i}(1 - \widehat{\mu}_{\mathrm{BE}}^{\,i})}{n_j^i}\right). \quad (8)$$

Hence, the likelihood of a given VAF observation from the $M_v$ model can be calculated as:

$$P\left(x_{\mathrm{BE}\,j}^{\;i}|M_v\right) = N\left(x_{\mathrm{BE}\,j}^{\;i}; \widehat{\mu}_{\mathrm{BE}}^{\,i}, \widehat{\sigma}_{\mathrm{BE}}^{\,i\,2}\right), \widehat{\sigma}_{\mathrm{BE}}^{\,i} = \sqrt{\frac{\widehat{\mu}_{\mathrm{BE}}^{\,i}(1 - \widehat{\mu}_{\mathrm{BE}}^{\,i})}{n_j^i}}. \quad (9)$$

Overall, the log ratio of the probabilities for $M_v$ and $M_0$ is calculated as described below, and positions with $S^i > 0$ are called as variant candidates.

$$S^i = \log\left(\frac{\phi_v \prod_{j=1}^{k} N\left(x_{\mathrm{BE}\,j}^{\;i}; \widehat{\mu}_{\mathrm{BE}}^{\,i}, \widehat{\sigma}_{\mathrm{BE}}^{\,i\,2}\right)}{\phi_{\mathrm{BE}} \prod_{j=1}^{k} \mathrm{Exp}\left(x_{\mathrm{BE}\,j}^{\;i}; \lambda_{\mathrm{BE}}\right)}\right). \quad (10)$$

**Performance assessment with replicates.** Since no systematic method has been established to utilize replicates for variant detection, the most straightforward approaches (intersection, union, and BAM merge) were tested to evaluate the accuracy thereof. For samples A and B (0.5 and 1% VAFs), intersection and union sets of MuTect positive calls (calls with KEEP judgment) were tested for their performances on all platforms. Likewise, merged BAMs of replicates were analyzed by MuTect, and their positive calls were tested and compared. Conflict calls at the same position were discarded from intersection sets. Performance tests with RePlow and primitive approaches were achieved in all combinations of library triplicates ($X_{11}X_{21}$, $X_{21}X_{31}$, $X_{11}X_{31}$, and $X_{11}X_{21}X_{31}$) for every depth and platform. Average values with 95% confidence intervals are reported for the results with replicates.

All RePlow results in this article were obtained with the default parameter settings, regardless of sequencing platform. However, as a result of creating an excessive number of true mutations in the test-base data sets compared to their size of target regions, mutation rates were far beyond ordinary values ($8.05 \times 10^{-3}$, $8.60 \times 10^{-3}$, and $4.83 \times 10^{-3}$ for ILH, ILA, and ITA, respectively). Due to this intrinsic bias of mixture data, applying a typical mutation rate ($3 \times 10^{-6}$) to RePlow severely underestimated the amount of true mutations. Therefore, we used

actual mutation rates for performance tests to avoid such unrealistic distortions. We also applied actual mutation rates to MuTect for primitive approaches, and found that it worsened the overall accuracy by generating a larger number of false positives. We, thus, reported MuTect results with the default parameter settings, which showed better performances. Despite the underestimation, RePlow still showed the best overall performance with typical mutation rates, reflected in the extraordinarily low number of false positives (Supplementary Fig. 15).

To test performance in a multi-platform context, only the data sets with the highest depth of each platform were used (1000× for ILH and 10,000× for ILA and ITA). Due to differences in the designed targets between sequencing platforms, only overlapping regions were considered in the evaluation. Target regions and the answer set of true mutations were adjusted to each platform pair. As a result, 510, 483, and 500 true variants were selected for the answer set of ILHxILA, ILHxITA, and ILAxITA pairs. The performance of each method was measured for all nine combinations of library replicates from $X^y_{11}X^z_{11}$ to $X^y_{31}X^z_{31}$. Average values with 95% confidence intervals are reported as above.

For independent validation, Tru-Q7 reference standard (1.3% Tier, HD734, Horizon Dx, Cambridge, UK) was prepared and sequenced by ILH and ITA with the target coverages of 1000× and 10,000×, respectively. Two commercial cancer panels (SureSelect custom panel and Ion AmpliSeq cancer hotspot panel v2) that cover 83 and 50 cancer genes were used to mimic common experimental data. 35 cancer hotspot SNVs covered by both panels were selected as an answer set of true mutations. Library triplicates were made and sequenced for each platform, and performance tests were carried out for all combinations of triplicates as stated above. To verify the performance under the same conditions as in the actual analysis, a default mutation rate ($3 \times 10^{-6}$) was applied for all methods in the independent validation. Information for the target regions of cancer panels and selected hotspot mutations with observed allele frequencies in each replicate are listed in Supplementary Datas 3 and 4.

**Code availability.** The implemented program with the source code and a user-manual is available at https://sourceforge.net/projects/replow/

**Reporting summary.** Further information on experimental design is available in the Nature Research Reporting Summary linked to this article.

## Data availability
The NGS sequencing data of the test-base (36 samples from four spike-in materials and three matched controls in three sequencing platforms) and of the commercial reference genomes (six samples including triplicates and two matched controls in two cancer panels) have been deposited in the NCBI Sequence Read Archive with the accession code PRJNA517742 [https://www.ncbi.nlm.nih.gov/sra/PRJNA517742]. Deep targeted sequencing data of intractable epilepsy patients is available upon request to the corresponding author. The source data underlying Figs. 1b, 2c, 4a, b and Supplementary Figs. 1–3, 5–8, 10–13, and 15 are provided as a Source Data file. All other relevant data is available upon request.

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

## Acknowledgements

This research was supported by a grant from the Korea Health Technology R&D Project through the Korea Health Industry Development Institute (KHIDI), funded by the Ministry of Health & Welfare, Republic of Korea (Grant nos. HI15C1601 and HI14C1324 to S.K., and H16C0415 to J.H.L.).

## Author contributions

J.K., J.H.L., and S.K. initiated the idea. J.K. and S.K. developed the method. J.K., D.C., J.H.M., H.S., H.N., and S.K. worked on the data analysis and presentation. J.S.L. and J.H.L. prepared the material and conducted experimental validation. H.K. provided disease tissues. J.K., J.H.L., and S.K. prepared the manuscript. All authors read and approved the final manuscript.

## Additional information

**Competing interests:** The authors declare no competing interests.

