## [Peer Review File · Nature Communications]

Reviewers' comments:

Reviewer #1 (Remarks to the Author):

In this manuscript, Junho Kim and colleagues present a probabilistic method, RePlow, that is able to substantially improve detection sensitivity and specificity for DNA mutations at very low variant allele frequency (VAF), when sequencing datasets from two or more library replicates of the same underlying sample are available. This is a very impressive, detailed, carefully planned and executed study. The methods are carefully developed and clearly explained. Results are presented clearly and with statistical rigor. The improvements achievable with judicious use of library replicates are very compelling. The manuscript is extremely clearly written.

My only concern is the practicality of the approach: it is unclear how many cancer sequencing studies today utilize library replication, and whether or not the community will be open to doubling the cost of each experiment. In this regard, a useful analysis would be to compare accuracy (sensitivity/specificity) achievable from a single library replicate at a given depth of coverage to accuracy achievable by two library replicates, each sequenced to half that coverage.

A few additional, but minor comments:

1. It was not clear whether or not the RePlow probabilistic model has to be trained for each DNA preparation x library construction x sequencing technology combination, and how robust the model is with respect to changes in these technologies.
2. What are the resource requirements for and performance characteristics of RePlow i.e. what type of computer is required to run it, memory usage, runtime on realistic datasets, can it be run on whole-genome and whole-exome sequencing datasets?

Overall, this manuscript addresses a question that will be of interest to many cancer biologists, with a very high standard of analytical rigor.

Reviewer #2 (Remarks to the Author):

The authors present a new method, RePlow, to call somatic variants from sequencing data, introducing a number of methodological innovations on the framework of a previously described

somatic variant calling method, Mutect. The paper is generally well written and the supporting analysis provides reasonable evidence to show that this method improves false positive rates compared to current somatic variant calling tools when multiple library replicates are considered.

Major Comments:

1. Although the analysis presented in this publication focuses on the RePlow model's ability to leverage library replicates, the model seems to provide many method changes beyond representation of replicates, for instance 'on-the-fly' estimation of systematic errors stratified by mutation type. It would be useful for users to understand whether these methods provide any benefit for the analysis of single (non-replicate) libraries, and if this is not expected, demonstrating how the model performs in this degenerate case would still provide a useful control.

2. On line 246, one of the justifying statements for using library replicates is "VAFs of true variants are more concordant in replicates (blue vs. red dots in Fig. 3b)" but this statement is not quantified or supported by any systematic test.

3. The representation of M_0 (what I understood at first to be the non-variant model) in the methods could be further clarified. This model is first introduced on line 515 as "a reference model (M_0) that treats all mismatches as error calls.", but later it appears to be redefined to a background error model. For instance, in the prior definition on line 569, the implied state space for each site is (1) somatic variant sites (2) non-variant sites with systematic errors represented by the background error model (3) all events not represented by (1) and (2), such as sites with simple independent sequencing error. Despite this, only states (1) and (2) are represented in the prior. Likewise the likelihood definition for M_0 only addresses background error. Perhaps the original definition of M_0 could be more specifically defined to clarify what the log-odds score represents in the RePlow case?

4. The statement on line 560, that the term discussed can be regarded as "the number of positions that possesses mismatches caused by a background error.", appears to be inaccurate. The term $b_{\{BE\}^i_j}$ is based on a difference of expected values and should frequently be greater than 0 in the absence of background error due to simple sampling considerations. If this term, as defined, has been proven empirically useful then there may be no reason to change how it is computed, but it would be helpful to describe it instead as an empirical factor which is expected to positively correlate with background error, to help better highlight how this term is expected to behave, and identify areas of the model which could be made more precise in future iterations of the methods.

Minor Comments:

1. In the Data availability statement, there is no reference to the epilepsy patient data discussed in Figure 4d.

2. I could not find documentation to reproduce the RePlow analysis commands. Which version of RePlow was used and what command-line options were used? Ideally providing some command-line templates would help to improve reproducibility.

3. There are several aspects of the source code release that could be improved:
 - a. I could not find any statement of the software's license, so would ideally be added or made more visible.

 - b. There is no single point to acquire the full method distribution: I found (1) a tarball release of the code containing all the required compiled jars, R code and README, but not the RePlow source code (2) the RePlow java source code (only) under version control, without any build script, build instructions or README. If the source code is going to be released, providing the complete java source, R source, README, third-party java-archives and build script/build instructions under a single version control repository would be much more helpful to the community of scientific users.

 - c. In addition to build instructions, providing a very small demo dataset with the source release would benefit users by providing a very simple example and build verification.

Reviewer #3 (Remarks to the Author):

The paper describes an interesting new methods, called RePlow, that utilizes library level sequencing replicates to build a variant vs sequencing error model that can be used for calling variants with very low variant allele frequencies. This model allows for calling variants only seen in a small fraction of the reads, without producing substantially more false positives.

The authors argue that replicated sequencing runs will be trivial given how the costs of technology have been driven down. However in practice, for translation medicine, it is very likely that if additional sequencing runs can be afforded they would likely be utilized for analyzing other tumor biopsy portions or even other metastases. If the RePlow method could be applied in such cases,

where the samples are not library replicates but rather from the same individual, separated by some amount of time/distance, outlining the possibilities would make the paper much more impactful.

The authors seem to provide a good testing data set and an experiment based methodology for evaluating performance. The primary argument in justification of the method, and the added requirement of sequencing replicated samples, is based on the idea that all other methods are unable to provide this kind of fidelity. For this reason, the benchmark against the other methods is critical in justifying the methods requirements.

The benchmark against variant calling methods does cause some concern. It is described as using the various methods with default parameters, except in cases where a lower VAF threshold could be provided to increase sensitivity. However, there is no mention of additional post-filtering, which is common in modern sequence analysis pipelines. For example, the Broad's MuTect 'best practices pipeline' (<https://gatkforums.broadinstitute.org/firecloud/discussion/7512/broad-mutation-calling-best-practice-workflows>) includes numerous additional postfilters including:

- Panel of Normals: which removes sequencing seen repeatedly in a population, this not only removes common germline SNPs but common technical artifacts as well.
- FFPE Filter: which removes C>T and G>A mutations common in FFPE samples (which may not be an issue given the experimental design, but still worth noting)
- OxoG Filter: The dna oxidation damage event was in the paper as an example of the kind of artifact RePlow could detect. The OxoG filter could also deal with these issues.

Without also testing these existing methods for false positive filtering, its difficult to say that the lower false positive rate RePlow was able to achieve could not also be obtained with additional filtering on existing methods.

Reviewer 1

Remarks to the Author

In this manuscript, Junho Kim and colleagues present a probabilistic method, RePLOW, that is able to substantially improve detection sensitivity and specificity for DNA mutations at very low variant allele frequency (VAF), when sequencing datasets from two or more library replicates of the same underlying sample are available. This is a very impressive, detailed, carefully planned and executed study. The methods are carefully developed and clearly explained. Results are presented clearly and with statistical rigor. The improvements achievable with judicious use of library replicates are very compelling. The manuscript is extremely clearly written.

MAJOR QUESTIONS

Q1) My only concern is the practicality of the approach: it is unclear how many cancer sequencing studies today utilize library replication, and whether or not the community will be open to doubling the cost of each experiment. In this regard, a useful analysis would be to compare accuracy (sensitivity/specificity) achievable from a single library replicate at a given depth of coverage to accuracy achievable by two library replicates, each sequenced to half that coverage.

ANSWER to Q1)

We totally agree with the reviewer and so we have compared the performance of the two different numbers of library replicates (single, duplicate) with the same total depth of coverage (Fig.R1, see below). With duplicates, the depth of coverage for each replicate was a half of a single library, so that RePLOW with duplicates showed lower sensitivity than MuTect with a single library. However, RePLOW achieved far better precision with a very low false positive rate based on our probabilistic model, ultimately outperforming single libraries in F-score, the measure balancing precision and sensitivity, at a depth of 800X (400X per replicate) and greater in ILH. Therefore, generating replicates and utilizing RePLOW would be generally beneficial for the data with a total throughput of 800X and greater, and we think 400X per replicate is a reasonable requirement to ask for detecting mutations with ~1% VAF. The comparison with intersection clearly shows how well our method works for the same data. We reported this result with the supplementary figure (Fig. S11).

RePLOW's use of two library replicates is not expected to be at all cost prohibitive. The cost will not be doubled since both replicates can be sequenced simultaneously, resulting in no increase in cost for sequencing but just require some additional cost for library construction, which currently accounts about 20% of the total sequencing cost. At the same time, total sequencing costs themselves have dramatically decreased in the last decade and can be expected to continue to do so. We expect that the cost of sequencing itself will no longer act as a bottleneck in the near future so the overall cost of using RePLOW will raise no significant concern.

Figure R1. Performance assessment with the single and duplicate libraries for the same depth of coverage

MINOR QUESTIONS

Q1) It was not clear whether or not the RePlo probabilistic model has to be trained for each DNA preparation x library construction x sequencing technology combination, and how robust the model is with respect to changes in these technologies.

ANSWER to Q1)

We are sorry for the insufficient description of the probabilistic model estimation process. RePlo estimates the parameters of its probabilistic model on-the-fly from the given raw data. Hence, it doesn't require any trained values or prior information such as the sequencing platform used. This unique characteristic makes RePlo applicable to any combination of library preparation protocols and sequencing platforms without any additional parameter adjustments. All RePlo results in this paper were generated with the exactly same parameter settings, even for data sets from different sequencing platforms. We observed dramatic improvement for all types of data, which supports the robustness of our method regardless of the technologies used. We carefully revised the RePlo model section to clarify those points.

Q2) What are the resource requirements for and performance characteristics of RePlow i.e. what type of computer is required to run it, memory usage, runtime on realistic datasets, can it be run on whole-genome and whole-exome sequencing datasets?

ANSWER to Q2)

To analyze a 500X whole-exome sequencing dataset (~25Gb), RePlow takes ~9 hours per sample for the entire process with a single core and ~6.5Gb RAM usage. RePlow is not suitable for whole-genome datasets because it is designed to focus on detecting low-VAF mutations (~1% VAF), which requires deep enough depth for sequencing data, like 400X at least. We added this information to the software manual page. We appreciate the reviewer's helpful comment.

Overall, this manuscript addresses a question that will be of interest to many cancer biologists, with a very high standard of analytical rigor.

Reviewer 2

Remarks to the Author

The authors present a new method, RePlow, to call somatic variants from sequencing data, introducing a number of methodological innovations on the framework of a previously described somatic variant calling method, Mutect. The paper is generally well written and the supporting analysis provides reasonable evidence to show that this method improves false positive rates compared to current somatic variant calling tools when multiple library replicates are considered.

MAJOR QUESTIONS

Q1) Although the analysis presented in this publication focuses on the RePlow model's ability to leverage library replicates, the model seems to provide many method changes beyond representation of replicates, for instance 'on-the-fly' estimation of systematic errors stratified by mutation type. It would be useful for users to understand whether these methods provide any benefit for the analysis of single (non-replicate) libraries, and if this is not expected, demonstrating how the model performs in this degenerate case would still provide a useful control.

ANSWER to Q1)

As the reviewer pointed out, on-the-fly estimation of background errors also provides great benefits to remove false positives for the analysis of a single library (Fig.R2, see below). However, when the sequencing data contains a large number of background errors with VAFs similar to true mutations (e.g. ILA cases in the figure), the VAF cutoff becomes high enough to remove true mutations due to the high estimated error rate, resulting in significant decrease of sensitivity. With replicates of those error-rich data, RePlow can recover the removed true mutations based on the VAF concordance between replicates (Fig. 3b and variant model estimation in Method section), therefore showed far higher sensitivity. In addition, RePlow with replicates provides more sophisticated filtration of false positives, considering the error profile concordance; false positives with exceptionally high VAFs (therefore beyond the cutoff) from the analysis of single library can be subsequently filtered out when multiple replicates are considered together, based on their concordance of VAF and error probability. So all modules including the ones work well for the single library have benefits of replication. We added these contents to the software instruction page for the description of RePlow's single running mode.

Figure R2. Performance assessment of RePlow with the single and duplicate mode

Q2) On line 246, one of the justifying statement for using library replicates is “VAFs of true variants are more concordant in replicates (blue vs. red dots in Fig. 3b)” but this statement is not quantified or supported by any systematic test.

ANSWER to Q2)

We are sorry for the missing statistics to support the given argument. We utilized the cosine similarity from the $y=x$ line for the concordance measure, not to be affected by the absolute value of VAF. We have compared the distribution of cosine similarity for true variants and false positive errors, and achieved statistically significant difference between them (Fig. R3, Kolmogorov–Smirnov (KS) test, $P < 2.2 \times 10^{-16}$, $P = 9.2 \times 10^{-13}$, $P = 6.7 \times 10^{-9}$ for ILH, ILA, ITA). We added this result to the main text with the additional supplementary figure (Fig. S9).

Figure R3. Comparison of VAF concordance between the true and false positive calls

Q3) The representation of M_0 (what I understood at first to be the non-variant model) in the methods could be further clarified. This model is first introduced on line 515 as “a reference model (M_0) that treats all mismatches as error calls.”, but later it appears to be redefined to a background error model. For instance, in the prior definition on line 569, the implied state space for each site is (1) somatic variant sites (2) non-variant sites with systematic errors represented by the background error model (3) all events not represented by (1) and (2), such as sites with simple independent sequencing error. Despite this, only states (1) and (2) are represented in the prior. Likewise the likelihood definition for M_0 only addresses background error. Perhaps the original definition of M_0 could be more specifically defined to clarify what the log-odds score represents in the RePflow case?

ANSWER to Q3)

We agree with the reviewer that we need to clarify the description of M_0 . As the reviewer points out, our reference model accounts for both background and sequencing errors and tries to eliminate both from the variant candidate list. This is accomplished in two steps, which explains why only the background errors are considered in the prior and likelihood definition. In this paper, we have argued that background errors that are generated during the library preparation step cause a significant number of low-level false positive candidates, and therefore need to be removed. Since there is no way to directly measure the amount of background errors for a given observation, we estimated it by subtracting the portion that can be explained by sequencing errors from all mismatches, which portion had previously been determined using base call quality scores. For those residuals unexplained by sequencing errors (we defined it as the mismatch over-representation score (MOS) score), our error model tries to calculate the probability that the given *unexplained residual* originated from the background error. Please note that both error and variant models in our methods use the adjusted VAF ($X_{BE_j}^i = \text{MOS}_j^i / n_j^i$), not the raw VAF, to only consider those unexplained residuals. In other words, among the three possibilities for a given mismatched site ((1) somatic variant site, (2) background error, (3) sequencing error), we first eliminate the portion that can be explained by (3) and try to discriminate (1) from (2) using the error and variant models with the remaining unexplained residuals. If a given site originated from sequencing errors, then most mismatches will be explained by (3) based on their base call quality, and therefore will result in very small MOS score. A small MOS score results in a very low adjusted VAF regardless of the raw VAF, and will not be considered to be a true variant. Nevertheless, our probability models only address background errors in the prior

and likelihood definition, yet they in fact account for both types of errors because they utilize the adjusted VAF, having already accomplished the preliminary step of excluding the portion that can be identified as sequencing errors. To clarify the model definition, we have carefully revised the *Error model estimation* section and provided a more adequate description. We again thank the reviewer for the question.

Q4) The statement on line 560, that the term discussed can be regarded as “the number of positions that possesses mismatches caused by a background error.”, appears to be inaccurate. The term $b_{BE_j}^i$ is based on a difference of expected values and should frequently be greater than 0 in the absence of background error due to simple sampling considerations. If this term, as defined, has been proven empirically useful then there may be no reason to change how it is computed, but it would be helpful to describe it instead as an empirical factor which is expected to positively correlate with background error, to help better highlight how this term is expected to behave, and identify areas of the model which could be made more precise in future iterations of the methods.

ANSWER to Q4)

We appreciate the reviewer’s comment. As the reviewer exactly pointed out, the term $b_{BE_j}^i$ itself is not equivalent to “the number of positions that possesses mismatches caused by a background error”, because the value can be >0 anytime even without the background error. And the suggestion is absolutely right. Therefore, we revised the description of the term as an empirical estimate of the amount of background errors, at which a larger value means more background errors in the sample.

MINOR QUESTIONS

Q1) In the Data availability statement, there is no reference to the epilepsy patient data discussed in Figure 4d.

ANSWER to Q1)

We apologize for omitting reference to the data from intractable epilepsy patients. These sequencing data sets are available upon request to the corresponding author. We added this statement to the Data availability section.

Q2) I could not find documentation to reproduce the RePlow analysis commands. Which version of RePlow was used and what command-line options were used? Ideally providing some command-line templates would help to improve reproducibility.

ANSWER to Q2)

We have updated all RePlow results with version 1.1, which is the most recent version released in our repository. We have used default parameter settings regardless of the sequencing platform, with the sole exception of the mutation rate parameter with the test-base dataset as we had described in Methods section. In this revision, we have added the exact command line we had used for our analysis in the supplementary materials.

Q3) There are several aspects of the source code release that could be improved:

- a. I could not find any statement of the software’s license, so would ideally be added or made more visible.
- b. There is no single point to acquire the full method distribution: I found (1) a tarball release of the code containing all the required compiled jars, R code and README, but not the RePlow source code (2) the RePlow java source code (only) under version control, without any build script, build instructions or README. If the source code is going to be released, providing the complete java source, R source, README, third-party java-archives and build script/build instructions under a single version control repository would be much more helpful to the community of scientific users.
- c. In addition to build instructions, providing a very small demo dataset with the source release would benefit users by providing a very simple example and build verification.

ANSWER to Q3)

We appreciate the reviewer's helpful comments. We now have added the statement of license in the software distribution package. Our software is freely usable for academic, non-commercial research purpose. In addition to the compiled tarball release, we have updated our repository to provide the source codes with build instructions. We have also added a small demo dataset for build verification, as the reviewer suggested. We again thank the reviewer for kind recommendations.

Reviewer 3

Remarks to the Author

The paper describes an interesting new methods, called RePLOW, that utilizes library level sequencing replicates to build a variant vs sequencing error model that can be used for calling variants with very low variant allele frequencies. This model allows for calling variants only seen in a small fraction of the reads, without producing substantially more false positives.

MAJOR QUESTIONS

Q1) The authors argue that replicated sequencing runs will be trivial given how the costs of technology have been driven down. However in practice, for translation medicine, it is very likely that if additional sequencing runs can be afforded they would likely be utilized for analyzing other tumor biopsy portions or even other metastases. If the RePLOW method could be applied in such cases, where the samples are not library replicates but rather from the same individual, separated by some amount of time/distance, outlining the possibilities would make the paper much more impactful.

ANSWER to Q1)

We appreciate the reviewer's thoughtful comment. In the case of true mutations that are shared between two different sites from the same tumor or individual, we can assume this pair to be *biological replicates*. We think that applying RePLOW to these samples can be also beneficial to accurately detect shared mutations by discriminating low-level systematic errors that often disguise the true mutation profiles. And this approach would be also helpful in detecting a second-hit mutations in tumors that are expected to have very low allelic fractions or to analyze the lineage and clonality. Based on the helpful comment, we have now added such potential values to Discussion.

Q2) The authors seem to provide a good testing data set and an experiment based methodology for evaluating performance. The primary argument in justification of the method, and the added requirement of sequencing replicated samples, is based on the idea that all other methods are unable to provide this kind of fidelity. For this reason, the benchmark against the other methods is critical in justifying the methods requirements.

The benchmark against variant calling methods does cause some concern. It is described as using the various methods with default parameters, except in cases where a lower VAF threshold could be provided to increase sensitivity. However, there is no mention of additional post-filtering, which is common in modern sequence analysis pipelines. For example, the Broad's MuTect 'best practices pipeline'

(<https://gatkforums.broadinstitute.org/firecloud/discussion/7512/broad-mutation-calling-best-practice-workflows>) includes numerous additional postfilters including:

- Panel of Normals: which removes sequencing seen repeatedly in a population, this not only removes common germline SNPs but common technical artifacts as well.
- FFPE Filter: which removes C>T and G>A mutations common in FFPE samples (which may not be an issue given the experimental design, but still worth noting)
- OxoG Filter: The dna oxidation damage event was in the paper as an example of the kind of artifact RePLOW could detect. The OxoG filter could also deal with these issues.

Without also testing these existing methods for false positive filtering, it's difficult to say that the lower false positive rate RePlow was able to achieve could not also be obtained with additional filtering on existing methods.

ANSWER to Q2)

We understand the reviewer's concern that the current lower false positive rate obtained from RePlow might also be achieved by conventional post-filtering steps. To address this concern, we have applied several common post-filtering steps to the results of MuTect to confirm that false positives are not removed to the same dramatic extent as with our RePlow method. We have tested four independent post-filtering steps including the three suggested by the reviewer: Panel of Normal (PoN), read pair orientation (accounts for both FFPE and OxoG artifacts), and FFPE filters (remove low-level NCG>NTG artifacts) (Fig. R4, see below). We had also shown the receiver operating characteristic (ROC) analysis of common features for variant filtering including base call quality, mapping quality, number of per-read mismatches, and indel proximity, to additionally support that most features that had been used for variant filtering were not able to mitigate excessive false positive rates (Fig. S4). We first looked for implemented tools or descriptions to address those post-filtering steps. Since there is no publically available PoN, we constructed our own PoN using 2,520 whole-exome sequencing data from the 1000 Genomes project, following the instruction from the GATK best practice workflow to remove every variant that has been detected in ≥ 2 normal individuals^{1,2}. Note that all true positive sites in our test-base data sets are common SNPs that are easily filtered out by PoN, so we exempted all true sites from the PoN filter. In this way, the PoN filter suffers no sensitivity loss, therefore may show too optimistic results by removing the risk of using PoN at all. OxoG artifacts has been well known to show a large imbalance between read pair orientations^{3,4}. To apply a read pair orientation filter, we utilized the computational filter designed by Costello et al⁴, a trained model considering read pair orientation bias and LOD_T values from MuTect to effectively filter out OxoG artifacts. Note that this filter could not be applied in ITA, because it is single-end read data from Ion Proton. We tested two more filters, FFPE and low-confidence filters, from the ngs-filters module developed by Memorial Sloan Kettering Cancer Center (<https://github.com/mskcc/ngs-filters>). The FFPE filter removes NCG>NTG candidates (C>T substitution at CpG sites) with the $VAF < 0.1$, and the low-confidence filter removes the candidate if it fulfills either one of followings: i) alternative count > 1 in the matched normal, ii) total depth < 20 in the disease sample, iii) alternative count < 4 in the disease sample. We described the details of used external filters in the Supplementary Methods.

The results showed that post-filtering steps did not effectively reduce the false positive rate (figure below), with the sole exception of the read pair orientation filter in ILH by the removal of OxoG artifacts, which we had already reported in the manuscript with Fig. S4. Most filters accompanied substantial sensitivity drops, which represents that false positives have similar characteristics to those of true low-level mutations and therefore could not be distinguished by those filters. The PoN filter had almost no effect compared to the results without any additional filters. Please remind that we only considered high-quality regions (only exons with average mapping quality ≥ 58 , containing only one variant with a read depth ≥ 20 , indel-free, containing clipped reads < 5 , see Methods) when we design our test-base data sets to avoid the influence of common systematic artifacts. Hence, the major source of false positives in our test datasets may not be common systematic artifacts and therefore most post-filtering steps, especially for PoN, showed less effective results than those of generally expected.

The reviewer may think that we constructed the PoN based on wrong data sets (totally different control sets from the test-base dataset) and therefore will have good results with the appropriate PoN. We admit that the constructed PoN will be totally different if the control samples have deep enough sequencing depth, although it is very difficult (or almost impossible) for now to get such multiple high-depth control datasets. However, even with the high-depth controls, we are skeptical to have good results due to the large fraction of the target regions with false positives in our test data sets. For example, at 10,000x in ILA, ~50% of target regions showed false positive calls despite their high quality (Fig. S7, note that they are high-confident variant candidates from MuTect with $LOD_T > 6.3$ and enough numbers of variant supporting reads, and about 80% of target regions had at least one read with the alternative allele). At the same time, those errors also accumulated at the ~30% of true

mutations sites and formed triallelic sites, preventing the variant detection by MuTect and resulting sensitivity loss. These results indicate that even if we can obtain an appropriate PoN from high-depth normal data, the PoN will cover the most genomic regions and cause a significant drop of sensitivity with the current filtering-based approach. The probabilistic model with enough numbers of high-depth control data sets will help to address this problem in the future. We reported the results of post-filtering steps with the supplementary figure (Fig. S5).

Figure R4. The benchmarks of conventional post-filtering steps from the test-base data (1% VAF)

Again, we thank all the reviewers for their careful reading of the manuscript and valuable comments.

References

1. DePristo, M.A. *et al.* A framework for variation discovery and genotyping using next-generation DNA sequencing data. *Nat Genet* **43**, 491-8 (2011).
2. Van der Auwera, G.A. *et al.* From FastQ data to high confidence variant calls: the Genome Analysis Toolkit best practices pipeline. *Curr Protoc Bioinformatics* **43**, 11 10 1-33 (2013).
3. Chen, L., Liu, P., Evans, T.C., Jr. & Ettwiller, L.M. DNA damage is a pervasive cause of sequencing errors, directly confounding variant identification. *Science* **355**, 752-756 (2017).
4. Costello, M. *et al.* Discovery and characterization of artifactual mutations in deep coverage targeted capture sequencing data due to oxidative DNA damage during sample preparation. *Nucleic Acids Res* **41**, e67 (2013).

REVIEWERS' COMMENTS:

Reviewer #1 (Remarks to the Author):

The authors satisfactorily addressed all of this reviewers's questions and comments.

Please note that although Reviewer #2 doesn't have remarks to the author, in his/her remarks to the editor, Reviewer #2 says he/she is satisfied with all answers from the authors.

Reviewer #3 (Remarks to the Author):

The additions to the paper help to eliminate the concerns about the justification of the method and the comparisons to other systems.